# Position : il faut donner du sens aux listes de lectures pour les jeunes chercheurs

Yannis Chupin[1]    Julien Aubert-Béduchaud[1]    Florian Boudin[2]    Richard Dufour[1]

(1) Nantes Université, École Centrale Nantes, CNRS, LS2N, UMR 6004, F-44000 Nantes, France

(2) Inria, LS2N, Nantes Université, France

`yannis.chupin@etu.univ-nantes.fr, julien.aubert-beduchaud@univ-nantes.fr,`
`richard.dufour@ls2n.fr, florian.boudin@inria.fr`

## RÉSUMÉ

Face à l'accroissement constant du nombre de publications scientifiques, accentué par le développement des grands modèles de langage (LLM), l'accès à la connaissance devient un défi majeur, particulièrement pour les chercheurs novices. Si les moteurs de recherche et les systèmes de recommandation isolent efficacement des articles pertinents, ils échouent à organiser ces ressources en parcours d'apprentissage cohérents. Cet article dresse un état de l'art de la génération automatique de listes de lecture, conçues non comme de simples agrégats de documents, mais comme des ensembles structurés visant une transmission efficace des connaissances. Les travaux scientifiques actuels cherchent à modéliser les dépendances et la progression pédagogique entre articles. En soulignant les limites de ces approches, notamment leur manque d'informativité, nous posons cinq critères pour répondre au besoin d'informativité et comparons cinq tâches à ces critères pour orienter la recherche sur cette thématique.

## ABSTRACT

**Addressing the lack of informativeness in reading lists intended for junior researchers**

With the constant surge in scientific publications, further accelerated by the development of Large Language Models (LLM), accessing knowledge has become a major challenge, especially for novice researchers. While search engines and recommendation systems effectively identify relevant articles, they often fail to organize these resources into cohesive learning pathways. This paper explores the automatic generation of reading lists, framed not as mere aggregates of documents, but as structured sets designed for the effective transmission of knowledge. We examine how current research attempts to model dependencies and pedagogical progression between articles. By highlighting the limitations of these approaches, particularly their lack of informativeness, we propose five criteria to bridge this gap and evaluate five tasks against these criteria to guide future research in this area.

MOTS-CLÉS : Listes de lectures scientifiques, génération, informativité, article de position, TAL.

KEYWORDS: Scientific reading lists, generation, informativeness, position paper, NLP.

# 1 Introduction

La communauté scientifique fait face à une croissance continue du nombre de publications. Entre 2012 et 2022, le volume annuel de publications scientifiques a augmenté de plus d'un million d'articles, passant de 2,1 à 3,3 millions (World Bank, 2024). De ce fait, connaître son domaine devient de plus

en plus difficile. En somme, la veille scientifique, la lecture approfondie et, plus largement, la montée en compétences sur un sujet se trouvent entravés par l'apparition soutenue de nouveaux articles. Le développement des grands modèles de langage (LLM), en facilitant l'écriture de ces articles, augure l'intensification de ce phénomène.

Cette évolution affecte tout particulièrement les publics novices pour le domaine. Il peut s'agir d'étudiants en master, de doctorants en début de thèse ou encore de chercheurs confirmés qui se tournent vers une nouvelle thématique. Ces profils partagent souvent la même difficulté : savoir par quel article commencer pour découvrir un domaine, et ce, sans disposer d'une vision claire des travaux les plus adaptés à leurs besoins d'apprentissage (Wang *et al.*, 2010). En recommandation, ces utilisateurs sont qualifiés de nouveaux utilisateurs (*cold start users*). Ils représentent un enjeu spécifique, car les algorithmes de recommandation ne peuvent pas utiliser leur historique (inexistant). Leur faible familiarité avec le domaine rend également plus difficile l'identification d'articles pouvant servir de point d'entrée pertinent (Chen *et al.*, 2025).

Plusieurs solutions existent pour faciliter l'apprentissage d'un domaine scientifique. Les moteurs de recherche académiques permettent de trouver des publications pertinentes, mais leurs critères de sélection demeurent souvent opaques. Ils ne permettent pas non plus d'organiser l'enchaînement des lectures, ce qui limite leur utilité pour un lecteur novice. Les revues de littérature offrent des synthèses précieuses, mais elles sont coûteuses à produire, peu nombreuses à l'échelle des besoins, et deviennent rapidement incomplètes. Les tutoriels, souvent proposés dans le cadre de conférences scientifiques, répondent souvent mieux aux besoins de nouveaux entrants dans un domaine, mais ils sont encore plus rares. Par exemple, la librairie digitale de ACM recense 48 267 tutoriels, contre 610 620 revues de littérature [1]. Enfin les capacités de synthèse des LLM montrent leur potentiel pour guider l'exploration de la littérature, mais plusieurs limites restent à étudier, notamment les hallucinations (Sakai *et al.*, 2026), les biais, par exemple selon la langue (Datta *et al.*, 2026), ainsi qu'un accès parfois décalé aux ressources les plus récentes disponibles en ligne.

Dans ce contexte, une direction prometteuse pour assembler le bagage scientifique d'un nouveau chercheur dans un domaine consiste à s'intéresser aux *listes de lecture scientifiques* (LLS). Une LLS vise à sélectionner un ensemble d'articles pertinents pour découvrir un sujet, tout en organisant leur lecture selon une certaine progression (Ekstrand *et al.*, 2010 ; Jardine, 2014 ; Aubert-Béduchaud *et al.*, 2025). Une telle approche répond à un besoin concret et souvent sous-estimé : non seulement savoir quels articles lire, mais aussi dans quel ordre les lire pour faciliter la compréhension. Cette direction reste toutefois encore peu étudiée. Par ailleurs, assembler une LLS ne se limite pas à trouver des articles pertinents. Il faut aussi tenir compte des dépendances entre travaux, du niveau de difficulté des articles, des notions préalables requises et des différents points d'entrée possibles dans un domaine. Une LLS représente donc plus qu'un amalgame d'articles, c'est un ensemble structuré ayant un but de transmission efficace des connaissances, d'où la complexité de cette tâche.

À l'occasion de ce travail, nous dressons un état de l'art de la génération de LLS. Nous discutons ensuite des limites des approches actuelles en observant que les listes générées actuellement manquent d'informativité pour les jeunes chercheurs. Face à ce constat, nous posons trois types de tâches répondant au besoin d'expliciter les LLS. Nous les confrontons ensuite à cinq critères d'informativité que nous définissons. Ce travail a pour objectif de faciliter l'exploration et l'apprentissage à partir de la littérature scientifique pour les jeunes chercheurs.

---

1. Chiffres obtenus en avril 2026 en cherchant respectivement "survey" et "tutorial" sur https://dl.acm.org/

## 2 Génération automatique de listes de lecture scientifiques

Cette section définit la tâche de génération de LLS, non comme une simple recommandation, mais comme la construction d'un parcours d'apprentissage cohérent. Nous y formalisons les enjeux de sélection et d'ordonnancement des articles (section 2.1), avant d'analyser les méthodes actuelles (section 2.2), les jeux de données disponibles (section 2.3) et les limites de leur évaluation (section 2.4).

### 2.1 Formalisation de la tâche

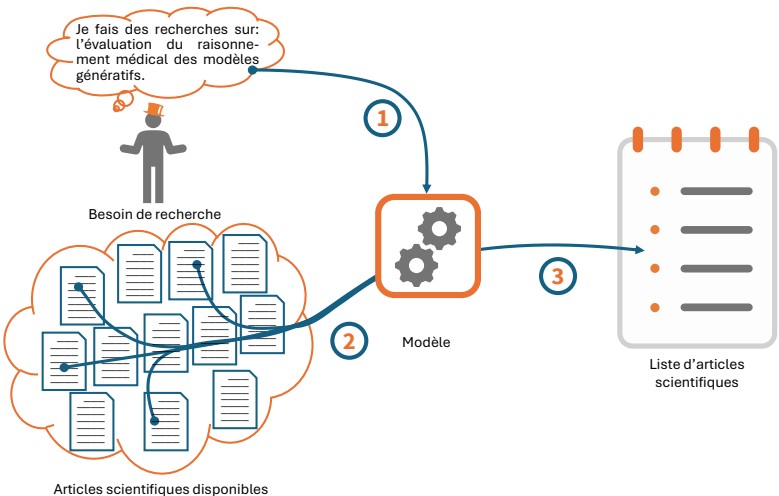

FIGURE 1 – Visualisation schématique de la génération de LLS.

Générer une LLS est avant tout une tâche de recherche d'information. Elle consiste à rassembler un sous-ensemble d'articles de taille restreinte [2] issu d'une collection scientifique (Jardine, 2014). Ces articles sont sélectionnés en fonction de leur pertinence par rapport au domaine ciblé (au moyen d'une requête), sont souvent formulés sous forme de mots clés, et ont pour mission d'être à la fois représentatifs du domaine visé tout en étant cohérents du point de vue des connaissances.

Là où la recherche d'information classique, illustrée par le classement par pertinence de Google Scholar (Beel & Gipp, 2009), traite chaque document de manière isolée, la constitution d'une LLS repose sur une logique d'ensemble. Ici, la pertinence ne s'évalue pas article par article, mais par la capacité de la sélection à appréhender les problématiques soulevées par le domaine ciblé de manière la plus complète et la moins redondante possible.

Le but des LLS est avant tout d'aider les novices à progresser dans leur compréhension. De ce fait, à la tâche de rassembler les articles pertinents se rajoute celle de présenter les articles de manière structurée pour guider le lecteur des articles les plus introducteurs vers les plus spécialisés. Cet ordre introduit est parfois qualifié de pédagogique (Gordon *et al.*, 2017) ou de chaîne de prérequis (Ding *et al.*, 2022; Fabbri *et al.*, 2018) dans la littérature, et renvoie à l'existence d'un chemin de lecture idéal pour parcourir les articles retenus.

---

2. Les listes de lecture scientifique contiennent souvent 20 articles maximum (Jardine, 2014; Figueira *et al.*, 2019).

## 2.2 Méthodes existantes

Pour traduire ce « chemin idéal » en critères algorithmiques, la littérature a exploré trois approches principales visant à ordonner les articles de la manière la plus pédagogique possible. Les premiers travaux (Ekstrand *et al.*, 2010; Wang *et al.*, 2010) ont privilégié la centralité structurelle. Par exemple, Ekstrand *et al.* (2010) proposent d'identifier les travaux fondateurs d'un domaine en appliquant l'algorithme *PageRank* (Page *et al.*, 1999) aux réseaux de citations. Cette méthode permet de dégager les références incontournables par leur autorité scientifique. Toutefois, elle tend à privilégier la popularité des articles plutôt que la progression logique des concepts.

Pour affiner cette sélection, une approche par graphe de concepts a été introduite par Gordon *et al.* (2017). En s'appuyant sur la similarité sémantique et l'analyse de l'entropie entre thématiques, ce modèle ordonne les articles selon un principe de prérequis pour constituer un graphe des concepts (Gordon *et al.*, 2016). Plus concrètement, le modèle utilise la théorie de l'information pour identifier les concepts « sources » (fondations) et « cibles » (spécialisations). Ensuite, il extrait le chemin le plus court au sein du graphe pour obtenir une LLS ordonnée et cohérente. Bien que plus informative, cette méthode impose une contrainte logistique majeure : elle nécessite l'accès au texte intégral des articles pour en extraire la structure interne, limitant ainsi son application à grande échelle ou dans des contextes interdisciplinaires.

Plus récemment, Ding *et al.* (2022) ont déplacé l'enjeu vers la génération de « chemins de lecture » (*Reading Path Generation*). En exploitant la topologie du réseau de citations via un algorithme essayant de trouver l'arbre de Steiner pondéré (Segev, 1987). Cette approche trouve le plus petit arbre des prérequis nécessaires à la compréhension de tous les articles considérés comme importants pour le domaine. Le modèle ne se contente plus d'indiquer « quoi lire », mais explicite ce qu'il faut connaître pour comprendre chaque facette du sujet cible au travers de la structuration en arbre.

## 2.3 Jeux de données

Peu de jeux de données recensant des LLS existent. À notre connaissance, cinq corpus ont été constitués pour le domaine scientifique. Parmi eux, trois sont accessibles publiquement. Ces ressources se divisent en deux approches méthodologiques : la collecte de listes expertes et la génération algorithmique. Leurs caractéristiques sont synthétisées dans la Table 1.

| Jeu de données | Domaine | # LLS | expert | Structurée | Type de structure |
|---|---|---|---|---|---|
| Ekstrand *et al.* (2010) | Informatique | 220 | non | non | / |
| Figueira *et al.* (2019) | Informatique, Ingénierie | 1648[3] | non | oui | ordre de pertinence |
| SurveyBank (2022) | TAL, ML, IA | 9321[3] | non | oui | arbre de Steiner |
| Jardine (2014) | TAL | 8 | oui | non | / |
| ACL-rlg (2025) | TAL | 85 | oui | oui | sections thématiques |

TABLE 1 – Liste des jeux de données liés aux listes de lecture scientifiques, « expert » indique si les données sont réelles ou générées tandis que « structurée » indique si une structure notable est en place.

Tout d'abord, les listes générées ((Ekstrand *et al.*, 2010; Figueira *et al.*, 2019), SurveyBank) permettent un passage à l'échelle, mais posent des questions analytiques. En effet, l'ordonnancement n'y est

---

3. Compte le nombre de revues de littérature à partir duquel leur système construit des listes de lecture

pas aléatoire : il suit des critères d'optimisation où chaque nouvel article inséré en position $i$ doit voir ses prérequis couverts par les articles précédents. Cependant, nous analysons cette approche comme étant intrinsèquement imparfaite. Ces modèles conçoivent les prérequis sous le prisme d'un graphe de connaissances, alors que la LLS est, par essence, linéaire. Cette tension entre la structure multidimensionnelle du savoir et la linéarité du format « liste » constitue une limite majeure des approches actuelles : elles tentent de projeter une complexité de réseau sur un fil unique, risquant ainsi de briser certaines continuités logiques de l'apprentissage.

À l'inverse, les jeux de données issus d'experts (Jardine (2014), ACL-rlg (Aubert-Béduchaud *et al.*, 2025)) font face à un défi de volumétrie, le nombre de LLS restant très limité. D'un côté, ces LLS restent de simples collections plates d'articles. De l'autre, leur valeur réside dans la légitimité de l'expert : on peut émettre l'hypothèse que l'ordre proposé est pédagogiquement pertinent, même si cette intention n'est pas toujours explicitée. Une exception notable est le jeu de données ACL-rlg, qui est collecté au sein d'une structure spécifique. Contrairement aux listes « plates », celles-ci sont découpées en sections thématiques. Cette structuration est cruciale, car elle permet d'étudier comment rendre la littérature scientifique plus accessible aux jeunes chercheurs, en ne leur offrant pas seulement une liste, mais une véritable carte conceptuelle du sujet.

## 2.4   Mesurer la qualité des listes de lectures scientifiques

La qualité des systèmes de recherche d'information classique repose sur la pertinence de chaque proposition vis-à-vis du *besoin de recherche* (requête initiale). L'évaluation d'une LLS relève toutefois d'un objet pédagogique multidimensionnel. Au-delà de la pertinence individuelle des articles, l'évaluation de leur qualité s'intéresse à la complémentarité des articles, de la complétude de la couverture thématique et de la cohérence du parcours proposé. À ce jour, il n'existe pas de métrique dédiée. La littérature adapte donc des outils issus de la recommandation et de la recherche d'information, bien qu'ils peinent à capturer la dimension structurelle de la tâche.

Pour évaluer la capacité d'un système à isoler les articles fondamentaux au sein d'une masse documentaire, les métriques de classification classiques constituent un premier jalon. La *précision $P@k$*, le *rappel $R@k$* et le *F1-score* permettent de quantifier l'adéquation entre la liste générée et une liste de référence (souvent établie par des experts). Cependant, ces mesures sont limitées, car elles traitent les documents comme des entités indépendantes. Pour pallier cela, la littérature mobilise *Mean Average Precision (MAP)* (Li *et al.*, 2024), qui évalue la capacité du système à placer les articles les plus pertinents en début de liste. Pour un ensemble de requêtes $Q$, elle se définit par :

$$MAP = \frac{1}{|Q|} \sum_{q \in Q} \left( \frac{1}{R_q} \sum_{k=1}^{n} P@k \times rel(k) \right)$$

où $R_q$ est le nombre de documents pertinents pour la requête $q$, $P@k$ la précision au rang $k$ et $rel(k)$ une fonction indicatrice valant 1 si le document au rang $k$ est pertinent, 0 sinon. *Mean Average Recall (MAR)*, pour le rappel, se mesure de la même manière en remplaçant $P@k$ par $R@k$.

Le *Normalized Discounted Cumulative Gain (NDCG)* permet, contrairement aux approches *MAP* et *MAR* (Tamm *et al.*, 2021), d'utiliser des scores de pertinence granulaires, ce qui est particulièrement adapté pour modéliser une progression pédagogique, un ordre dans la LLS. Le score repose sur le Gain Cumulé Actualisé ($DCG_p$), qui pénalise de manière logarithmique la position d'un article en fonction de son score de pertinence $rel_i$ :

$$DCG_p = \sum_{i=1}^{p} \frac{2^{rel_i}-1}{\log_2(i+1)} \quad \text{et} \quad NDCG_p = \frac{DCG_p}{IDCG_p}$$

où $rel_i$ est la note attribuée à l'article au rang $i$. Afin de permettre la comparaison entre différentes LLS, ce score est normalisé par le gain idéal ($IDCG_p$), obtenu en calculant le DCG d'un ordonnancement parfait des articles par ordre décroissant de pertinence.

Bien que des métriques comme le $NDCG$ valorisent le placement des articles à fort impact en amont du parcours, elles échouent à capturer la fluidité pédagogique et la structure logique des connaissances. La nature non-linéaire du savoir et l'absence de "chemin de lecture unique" rendent la comparaison à une référence complexe, deux experts pouvant proposer des parcours divergents, mais valides. Dès lors, l'*évaluation humaine* demeure l'arbitre final pour évaluer la cohérence locale, la réponse aux problématiques soulevées par le domaine et la charge cognitive de l'enchaînement. À cette fin, les échelles de Likert semblent pertinentes, mais pour aller plus loin, Gordon *et al.* (2017) proposent de faire réordonner les LLS générées par des experts en respectant la chaîne des prérequis. L'écart avec la LLS originale est ensuite quantifié par la distance d'édition, permettant de mesurer précisément la pertinence de l'ordonnancement généré.

# 3 Plus que générer, il faut aussi expliquer

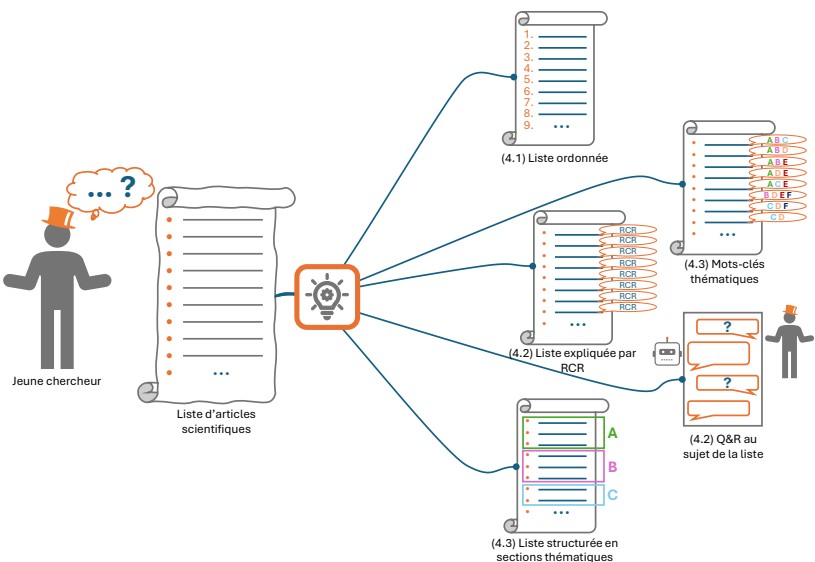

FIGURE 2 – Différentes approches pour rendre les listes plus explicites.

Nous avons donc vu que nous savons, au moins partiellement, générer des LLS et les évaluer. Mais nous pensons que c'est insuffisant puisque répondre au besoin de monter en compétence ne peut être satisfait par le seul contenu généré. En effet, pour un chercheur débutant, une simple sélection de titres ne suffit pas à construire un parcours d'apprentissage cohérent. Les travaux sur les attentes des étudiants soulignent d'ailleurs deux besoins majeurs. D'une part, ils recherchent une structure claire

et apprécient particulièrement les notes d'experts qui expliquent l'intérêt des articles (Kumara *et al.*, 2023). D'autre part, sans ces indications qui servent de guide, les jeunes chercheurs peuvent de se perdre dans les premiers articles (Siddall & Rose, 2014).

Le défi actuel est donc de passer d'une LLS automatique à un véritable support de transmission. Il ne s'agit plus seulement de proposer des articles, mais de permettre aux jeunes chercheurs de comprendre de quoi est constituée la LLS en ayant des connaissances encore limitées sur ce domaine. Pour répondre à ce besoin, nous disposons d'une LLS $L$ déjà assemblée, que ce soit par un expert ou un modèle. On peut la voir ici comme un ensemble d'articles $a$ tel que $L = \{a_0; a_1; ...; a_n\}$. Nous disposons aussi du *besoin de recherche* $B$ qui a motivé la création de la LLS. Il peut entre autres prendre la forme d'un titre, d'un sujet (de thèse, de stage) ou bien d'un prompt.

# 4 Plusieurs tâches visant l'informativité

Pour permettre au jeune chercheur d'optimiser sa lecture, nous cherchons à utiliser $B$ et $L$ introduits dans la section précédente pour rendre la LLS plus informative. À cette fin, nous présentons dans cette section des tâches qui permettent chacune d'expliquer une facette des LLS.

## 4.1 Ordonner les articles au sein de la liste scientifique

Ordonner une LLS la rend plus informative, car elle permet au lecteur de savoir par où commencer la lecture. Cette tâche cependant ne prend pas $B$ en compte pour fonctionner, puisqu'elle se contente d'assigner une position aux articles $a$ en fonction d'éléments le plus souvent issus de $a$. Dans la section 2.2 nous avons présenté les méthodes utilisées par les précédents travaux pour ordonner leurs LLS. Une première utilise l'ensemble du réseau des citations pour organiser les articles. L'intérêt est que les citations sont le reflet de la manière dont la communauté scientifique a construit la connaissance d'un domaine. Les citations indiquent donc implicitement comment progresser dans l'apprentissage d'un domaine.

Par ailleurs, une autre approche consiste à déterminer quelles notions sont nécessaires à la compréhension de chaque $a$ pour représenter ces articles dans un graphe. Cette approche met alors en place un système de prérequis pour organiser les articles en LLS ordonnées. L'avantage est que déterminer les prérequis repose sur des données disponibles. Cela rend aussi les choix explicables par les notions inférées. Un autre critère de classement est le type de publication. En effet, les articles tels que des revues de littérature sont par nature plus introductives que des articles présentant des innovations pointues. Ordonner une LLS présente cependant le défaut de ne donner que peu d'informations au lecteur : le choix tranché de mettre un des articles en premier ne permet au jeune chercheur ni de positionner ses connaissances face aux articles de la LLS ni d'expliquer ce qu'apporterait la lecture du premier article par rapport au deuxième. De plus, cette approche perd beaucoup en pertinence pour quelqu'un dont le besoin de recherche sort d'un domaine défini.

## 4.2 Utiliser le résumé automatique pour positionner les articles

Une manière de rendre le lien $B \times L$ explicite est de positionner chacun des articles par rapport au besoin de recherche, c'est-à-dire $\forall i \in [0; n], B \times a_i$. Pour expliciter ce positionnement, mettre

en place des résumés ciblés par requête (RCR) ([Liu *et al.*, 2024](#)) semble prometteur. Un RCR est à mi-chemin entre le Question-Réponse (Q&R) et le résumé classique de document. Il cherche à résumer un aspect ciblé du document qui est défini par une requête, $B$ dans notre cas.

**Les méthodes extractives** sont une approche éprouvée pour cette tâche. C'est ce que mettent en place les moteurs de recherche par exemple quand ils montrent les morceaux du document correspondant à la recherche effectuée. Le problème cependant est que les phrases extraites sont rarement représentatives de l'intérêt du document.

**Les méthodes abstractives** résolvent ce problème en reformulant les aspects sélectionnés du texte de manière synthétique. De cette manière, un jeune chercheur pourrait comprendre l'intérêt de l'article en une ou deux phrases courtes. Nous pensons ici qu'il est important que le résumé soit très court. En effet, rendre les LLS plus informatives a pour but de permettre au lecteur de se positionner efficacement. De ce fait, il est particulièrement important d'écrire le nécessaire en peu de mots.

**L'approche du Q&R** est encore meilleure pour répondre aux questions préliminaires et spécifiques d'un jeune chercheur sur la LLS. Elle permet de guider finement le lecteur sur l'intérêt de chaque article en répondant à des demandes spécifiques ([Deng *et al.*, 2024](#)). Mais, cette méthode a le désavantage de nécessiter des questions. Or un jeune chercheur se pose rarement les bonnes questions au début de ses recherches. De plus, cette approche est à la fois très coûteuse et difficilement évaluable.

## 4.3 Exposer la couverture thématique des articles

Mettre en évidence les thématiques abordées par les articles est un moyen pour le jeune chercheur de se positionner face aux articles proposés dans la LLS. Encore plus que précédemment cette tâche, qui s'apparente à la classification, fait face au défi de l'accessibilité des termes employés pour classifier les articles. En effet, exhiber des termes spécialisés [4] pour décrire la plus-value d'un article pourrait le rendre confus, puisqu'il n'est pas encore spécialiste du domaine.

**Les mots-clés** sont une manière d'exposer la plus-value des articles. Elle consiste à faire usage des mots-clés disponibles pour une majorité d'articles. En effet, les thématiques structurantes des articles y apparaissent clairement. Il se peut, cela dit, que seule une partie de l'article soit réellement intéressante du point de vue de $B$. Dès lors, générer les mots-clés pour prendre $B$ en compte semble être une piste digne d'intérêt. Cependant, il reste que le problème d'accessibilité des termes employés est exacerbé par la nature de mots-clés. Puisqu'aucun contexte ne les accompagne, une méconnaissance du terme suffirait au jeune chercheur pour mal s'orienter dans son cheminement de lecture. Les mots-clés sont faciles à évaluer et à obtenir (trouver dans des données ou générer), mais ils apportent un contexte insuffisant.

**La classification thématique** est au cœur des travaux de [Fabbri *et al.* (2018)](#) en explorant la classification des travaux par type de contribution (article court, long, revue...). Malheureusement cette ressource n'a pas pu être exploitée, puisque leur travail s'est concentré sur la création du jeu de données. Bien que l'on puisse imaginer qu'il soit intéressant de faire ressortir le type de l'article, ce n'est pas suffisant pour comprendre sa plus-value. Dès lors, il faut parler de la substance des articles scientifiques. Dans ce registre, [Aubert-Béduchaud *et al.* (2025)](#) en rassemblant des LLS a aussi collecté, pour certaines, **les sections** et sous-sections créées par les experts pour structurer leur liste. Cette démarche a un véritable intérêt, car elle fait deux choses : 1) elle regroupe des articles ayant

---

4. Fait référence ici à un terme technique utilisé dans un article utilisant des concepts états de l'art.

des points communs qui sont intéressants au vu de $B$, 2) au travers de l'intitulé des sections cette approche résume aussi ces points communs sous la forme d'une courte phrase souvent non verbale. Cette manière de faire rend la structuration de la liste claire et ouvre la voie à des expérimentations puisque des données existent.

# 5   Perspectives et discussions

## 5.1   Commencer par répondre au manque criant de données

Au vu du besoin d'informativité introduit dans la section 3, nous proposons des tâches issues du domaine du NLP pour y répondre. Cependant, il nous apparaît évident que le principal défi ne consiste pas à se saisir de ces pistes, mais bien de les évaluer. En effet, trop peu de données existent pour permettre de statuer sur la qualité d'une quelconque approche. Encore moins pour éventuellement affiner un modèle dédié. Il apparaît donc essentiel de concentrer les efforts de recherche sur ce point particulier.

## 5.2   Critères d'informativité des listes de lecture

Une fois le problème du manque de données surmonté, il reste à savoir vers quelle tâche, introduite dans la section 4, les recherches doivent se concentrer. La réponse est loin d'être évidente, car toutes apportent une vraie plus-value en termes d'informativité. Mais toutes les mettre œuvre pourrait s'avérer contre-productif. Par exemple, implémenter à la fois l'approche RCR abstractive et l'approche mots-clés donne une information similaire au lecteur, mais augmente considérablement le nombre de mots à lire au sein de la liste. Cette verbosité est contre-productive. Afin de choisir la meilleure combinaison, nous proposons cinq critères pour évaluer dans quelle mesure les tâches proposées répondent au besoin d'information des jeunes chercheurs.

1. **La concision** fait référence au nombre de mots employés pour rendre la liste plus informative. Ici, nous voulons que ces ajouts textuels soient les plus concis possible. En effet, des listes enrichies d'un contexte trop verbeux paraissent écrasantes (Siddall & Rose, 2014), ou trop longues (Kumara *et al.*, 2023).
2. **La structure** évalue dans quelle mesure la tâche facilite ou détériore la navigation au sein de la liste (Siddall & Rose, 2014; Kumara *et al.*, 2023; Centre for Teaching and Learning, 2024).
3. **L'accessibilité** statue de la difficulté des termes employés, car le public visé peut ne pas connaître ou avoir une mauvaise idée de la définition des termes spécialisés employés par les articles. L'objectif étant d'éviter de générer de la confusion chez les jeunes chercheurs (Kumara *et al.*, 2025).
4. **La contextualisation** est le critère qui évalue si la tâche proposée apporte des informations expliquant l'intérêt de lire $a$ au vu de $B$ (Centre for Teaching and Learning, 2024; Siddall & Rose, 2014).
5. **Le positionnement** des articles en fonction des autres, $L \backslash a \times a$, mesure si la tâche proposée permet d'observer en quoi les thématiques d'un article le rapprochent ou l'éloignent des autres articles de la liste (Centre for Teaching and Learning, 2024; Kumara *et al.*, 2023). Ce critère évalue aussi dans quelle mesure la tâche étudiée permet au lecteur de situer son bagage scientifique par rapport à l'article.

## 5.3 Trouver le meilleur compromis

Munis de ces cinq critères d'informativité, nous pouvons maintenant comparer les différentes tâches. La figure 3 montre comment les différentes tâches se différencient dans leur manière de communiquer l'information. La première leçon de ce comparatif est qu'aucune des tâches ne permet de complètement répondre au besoin d'informativité. Cependant, nous pensons que le couple Sections (thématiques) et Résumé (extractif ou abstractif) se détache. En effet, aucune d'elles n'est la pire option pour chaque critère et une des deux est toujours très bien placée pour chaque critère.

L'exception notable est le critère d'accessibilité, où Ordonner et Q&R sont meilleures. Respectivement, leur avantage est que le chercheur n'a pas besoin de connaître des termes pour comprendre un ordre, et s'il ne comprend pas un terme, il peut poser la question dans le cadre du Q&R. Pour autant, Ordonner échoue complètement sur le positionnement et la contextualisation ce qui le rend moins intéressant. Quant à Q&R même s'il a des qualités, la difficulté d'évaluer des LLM est connue or ces approches reposent en grande partie sur ces modèles quand ils sont interactifs (ce que nous évaluons ici). Enfin, on remarque que mots-clés est très similaire à Résumé, mais qu'il sous-performe en contextualisation, alors que c'est l'endroit où Sections a le plus besoin d'être complémenté.

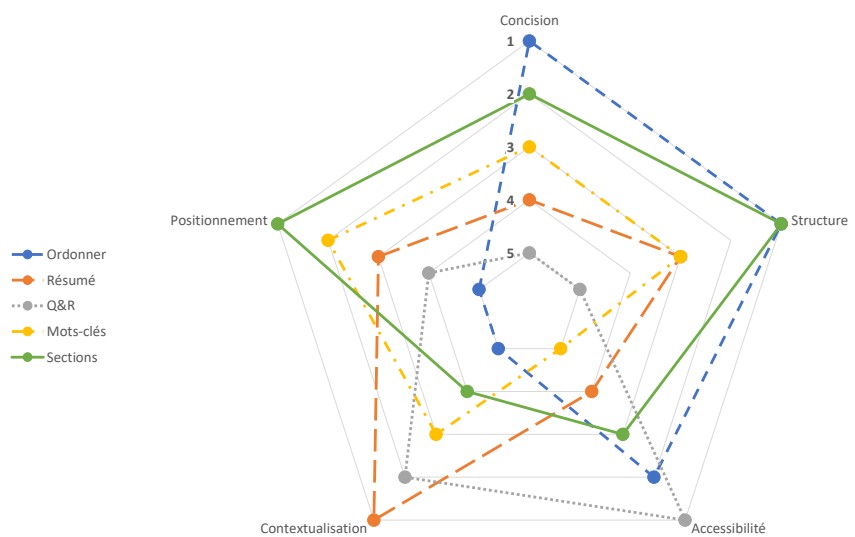

FIGURE 3 – Classement des différentes tâches à répondre aux cinq critères d'informativité d'une liste de lecture. 1 est le meilleur rang.

# 6 Conclusion

Dans cet article, nous avons expliqué que les jeunes chercheurs, en tant que lecteurs, ont besoin de listes de lecture qui explicitent leur constitution. Nous avons dès lors expliqué comment de nombreuses tâches pourraient répondre à ce besoin de clarté. Il en est ressorti qu'aucune de ces tâches ne répond complètement au besoin des jeunes chercheurs, pour différentes raisons. Nous pensons donc que combiner certaines d'entre elles est la meilleure manière de répondre au besoin, puisqu'elles

ne sont pas incompatibles. Cependant, il reste le défi de collecter les données qui permettraient d'évaluer quelque approche que ce soit. Cette poursuite de recherche est inévitable et nous souhaitons continuer sur cette voie.

# Remerciements

Nous remercions l'équipe TALN du laboratoire LS2N pour nous avoir accueillis, ainsi que les relecteurs anonymes pour leurs retours constructifs.

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
