# OpenReview forum: "Addressing the lack of informativeness in reading lists intended for junior researchers"
_ls2n.fr/CORIA-TALN/2026/Workshop/ARTS — ls2n CORIATALN 2026 Workshop ARTS Submission_

### Official Review · Reviewer_46wG · 2026-04-27

**Mode De Presentation:** Poster

**Confience:**

Oui

**Decision:**

Accepté

**Relecture:**

L'article est clairement dans les thématiques de l'atelier. Il décrit une tâche nouvelle, qui commence à faire l'objet de quelques travaux - peut-être pas assez pour faire l'objet d'une étude de la littérature, qui ne discute qu'une poignée de documents. L'article est dans l'ensemble assez bien rédigé, même si la structure d'ensemble et le cadrage sont très discutables: la section 3 est longue d'un paragraphe, et rate un peu la transition entre sections 2 et 4. Il me semble qu'il sera nécessaire de réorganiser l'ensemble pour mieux lier les premières sections (1 et 2) et les deux dernières (4 et 5).

Concernant le cadrage: les auteurices prennent pour acquis que la liste de lecture est le bon objet pour assister un chercheur junior à pénétrer un nouveau domaine. On pourrait penser, et c'est un peu ce que la seconde partie met en exergue, que ce qui est nécessaire ce sont (a) des concepts qui sont les objets sur lesquels travaillent les chercheurs; (b) les méthodes (formelles ou expérimentales, associées à des protocoles et des jeux de données); (c) les théories éventuellement contradictoire qui s'opposent et définissent les questions ouvertes ou en débat. Partant, on pourrait penser qu'il faudrait d'abord organiser une telle ontologie, puis associer des documents (articles, supports de cours, livres, tutoriels) qui permettront de se familiariser avec ces concepts. Cette manière de faire serait "explicable par design" -- et il doit exister une littérature épaisse sur cette question ou sur les questions reliées (par. ex: la génération automatique de parcours d'apprentissage dans des cadres plus scolaires). Les supports pédagogiques (manuels) sont souvent déjà structurés de cette manière et il est étrange de ne pas commencer par les discuter -- ne serait-ce que pour constater qu'il en existe sans doute trop peu.

Un second point un peu aveugle de la discussion est celle de la personnalisation: certains concepts peuvent être déjà connus ou maîtrisés, d'autres entièrement nouveaux. Il me semble que c'est également une question qui devrait figurer à l'agenda de recherche.

**Resume:**

Ce travail est une étude de la littérature sur la tâche de génération automatique de listes de lectures, destinées par exemple à fournir à des jeunes collègues désireux de prendre pied dans un nouveau domaine scientifique une liste ordonnée d'articles à lire, l'ordre de lecture garantissant une progression dans la compréhension des principaux concepts d'un domaine. Il s'agit d'une tâche originale, qui dépasse la génération automatique de bibliographies pour des études de littérature. Les auteurs présentent dans un premier temps la tâche, les méthodes proposées dans la littérature, les données existantes, et les questions d'évaluation. La seconde partie de l'article discute d'un besoin connexe: expliquer l'ordre de lecture pour rendre la liste plus informative. L'article se clôt sur une discussion et des perspectives.

---

### Official Review · Reviewer_4sgq · 2026-05-02

**Mode De Presentation:** Poster

**Confience:**

Oui

**Decision:**

Accepté

**Relecture:**

Exposition très claire des tâches et des critères, et du compromis à trouver entre celles-ci. Il sera intéressant de mettre ces critères à l’épreuve de jeux de données dans d’autres domaines que la linguistique computationnelle.

Quelques coquilles relevées :
-  2.1 : "et ont pour mission d’être à la fois représentatif du domaine visé tout en étant cohérent du point de vue des connaissances" => représentatif**s** et cohérent**s** (pluriel)
- 2.2 :
   - "En exploitant la topologie du réseau" => il manque un sujet et un verbe dans cette phrase (« Ils exploitent ») ou la rattacher à la phrase précédente à l’aide d’une virgule
   - "Le plus petit arbre des prérequis nécessaire à la compréhension" => si "nécessaire" se rapporte à "prérequis" alors il faut un "s" à "nécessaire**s**"
- 2.3 :
   - Légende du tableau 1 : "structure indique" => "structur**é**e indique" (pour être raccord avec le titre de la colonne, même si on comprend de quoi il s’agit)
   - Les 2 paragraphes après le tableau doivent être inversés : d’abord le paragraphe sur les listes d’expert, puis celui sur les listes générées (pour que le marqueur du discours « A l’inverse » soit plus logique)
- 4.1 : "Dans la section 2.2 nous présentons" => l’imparfait serait plus adapté ("nous présent**i**ons")
- 5.2 :
   - Pour le critère de concision, n’y a-t-il pas un qualificatif plus parlant pour caractériser les trop longues listes de Kumara et al. plutôt que "trop longues" ?
   - Pour le critère de positionnement : "permet d’observer, en quoi les thématiques d’un article, le rapproche ou l’éloigne des autres articles" => "permet d’observer en quoi les thématiques d’un article le rapproche**nt** ou l’éloigne**nt** des autres articles" (accord + enlever les virgules après "observer" et "article")
- 5.3
   - "ou Ordonner et Q&R sont meilleures" => "o**ù**"
   - "quand, ils sont interactifs" => virgule mal placée. A enlever ou à mettre avant « quand »
   - "mots-clés est très similaire à contextualisation, mais qu’il sous-performe en contextualisation" => je ne comprends pas en quoi mots-clés est similaire à contextualisation. S’agit-il d’une erreur ?

**Resume:**

Cet article de position met en avant le besoin de listes de lectures scientifiques organisées pour guider un jeune chercheur dans son appropriation d’un domaine scientifique. Il propose un état de l’art des méthodes de génération automatique de telles listes et présente 5 tâches permettant d’expliquer l’organisation de ces listes de lecture et 5 critères pour évaluer ces tâches et orienter la recherche.

---

### Decision · Program_Chairs · 2026-05-07

Accept (Poster)